# Extending the audiogram with loudness growth: The complementarity of electric and acoustic hearing in bimodal patients

**Lars Lambriks**○\*, **Marc van Hoof, Erwin George, Elke Devocht**○

Department of ENT/Audiology & School for Mental Health and NeuroScience (MHENS), Maastricht University Medical Centre, Maastricht, The Netherlands

\* lars.lambriks@mumc.nl

## Abstract

### Introduction

Clinically, recording hearing detection thresholds and representing them in an audiogram is the most common way of evaluating hearing loss and starting the fitting of hearing devices. As an extension, we present the loudness audiogram, which does not only show auditory thresholds but also visualizes the full course of loudness growth across frequencies. The benefit of this approach was evaluated in subjects who rely on both electric (cochlear implant) and acoustic (hearing aid) hearing.

### Methods

In a group of 15 bimodal users, loudness growth was measured with the cochlear implant and hearing aid separately using a loudness scaling procedure. Loudness growth curves were constructed, using a novel loudness function, for each modality and then integrated in a graph plotting frequency, stimulus intensity level, and loudness perception. Bimodal benefit, defined as the difference between wearing a cochlear implant and hearing aid together versus wearing only a cochlear implant, was assessed for multiple speech outcomes.

### Results

Loudness growth was related to bimodal benefit for speech recognition in noise and to some aspects of speech quality. No correlations between loudness and speech in quiet were found. Patients who had predominantly unequal loudness input from the hearing aid, gained more bimodal benefit for speech recognition in noise compared to those patients whose hearing aid provided mainly equivalent input.

### Conclusion

Results show that loudness growth is related to bimodal benefit for speech recognition in noise and to some aspects of speech quality. Subjects who had different input from the hearing aid compared to CI, generally gained more bimodal benefit compared to those patients whose hearing aid provided mainly equivalent input. This suggests that bimodal

**Data Availability Statement:** All relevant data are within the manuscript and its Supporting Information files. Speech outcomes are presented

in supporting S5 Table and original loudness data in supporting S1 Dataset.

**Funding:** A research grant from Advanced Bionics Inc. to Maastricht University Medical Centre (MUMC+) financially supported the work of the first (Lars Lambriks) and last author (Elke Devocht) in this investigator-initiated study. The funder had no role in study design, data collection and analysis, decision to publish, or preparation of the manuscript.

**Competing interests:** The authors have declared that no competing interests exist.

**Abbreviations:** HA, Hearing Aid; CI, Cochlear Implant; CIHA, Cochlear Implant and Hearing Aid worn together; ACALOS, Adaptive CAtegorical LOudness Scaling; NB, Narrowband; BB, Broadband; PTA, Pure Tone Average; IFFM, International Female Fluctuating Masker; CU, Categorical Units.

fitting to create equal loudness at all frequencies may not always be beneficial for speech recognition.

## Introduction

The functioning of the human auditory system depends on the amount of information that is delivered to both ears [1, 2]. Conventionally, the audiogram captures the threshold at which sounds can be detected across frequencies. We propose to extend the audiogram with an additional property: loudness growth. We argue that by integrating loudness growth, we gain a better estimate of the total available auditory information. In this study, we show the additional benefit of taking such an approach for patients who depend on two types of hearing modalities: electric and acoustic.

### Loudness growth

Typically, loudness growth for normal hearing listeners follows a function of the tone's intensity to the power 0.23 or an inflected exponential function [3, 4]. Hearing impaired listeners show large individual differences in loudness growth curves, but some common patterns have been identified. In subjects with classical loudness recruitment, loudness at threshold and at high levels is similar to that of normal hearing listeners, but more rapid loudness growth takes place in between [5].

To the best of our knowledge, loudness growth curves have mainly been measured for the diagnoses of loudness recruitment [6–8] and as a part of the HA fitting process [9, 10]. In clinical routine, audiometric thresholds remain the most influential audiological parameter to determine target gain and compression for hearing aid amplification. However, measuring loudness growth provides more information: minimal audible level, most comfortable level and its entire course in between [11]. Studies have shown that subjects with similar auditory dynamic ranges frequently show different courses of loudness growth [6, 12], emphasizing its distinctiveness from regular thresholds.

Few studies have examined the relationship between loudness growth and speech recognition directly. A study by Van Esch and Dreschler [13] showed that loudness recruitment, defined as the slope of the lower part of the loudness curve, was significantly related to speech recognition in fluctuating noise, but only accounted for an additional 3% of the explained variance above other auditory measures. Similarly, loudness recruitment has been found to account only for a small part of the explained variance on top of unaided Pure Tone Average (PTA) for speech recognition in noise [14]. Both studies concluded that loudness recruitment was related to poorer speech recognition. However, only unaided loudness growth was considered. It remains unclear how common characteristics of hearing devices, such as a hearing aid (HA) or a cochlear implant (CI), influence the relationship between loudness growth and speech recognition.

### Bimodal hearing

An increasing number of adult patients who qualify for a cochlear implant still have aidable residual hearing in the non-implanted ear [15, 16]. For these patients, wearing a contralateral HA often provides superior speech recognition than wearing CI alone [17]. Bimodal hearing refers to the situation where wearing a CI in one ear and a conventional hearing aid in the opposite ear (CIHA) outperforms making use of the CI alone. The difference between performance with CIHA and CI is defined as bimodal benefit and might be explained by two

mechanisms. First, the low-frequency residual hearing in the non-implanted ear provides complementary information since it contains fundamental frequencies of speech input and more temporal fine structure cues than electrical input [18]. Secondly, receiving input from two ears instead of one, provides access to binaural cues, and thereby facilitates several mechanisms (i.e. head shadow, squelch and summation) of which the importance in complex listening environments is well established [19, 20]. In bimodal subjects however, binaural cues can be limited due to auditory input originating from two different hearing devices (acoustic and electric) and limitations of residual hearing (acoustic) and mapping in the cochlea (electric).

**Bimodal benefit.** Bimodal hearing has been found to improve speech recognition, listening effort and sound localization in adult subjects [17, 21, 22]. Also, sound quality, which is often perceived as unnatural with CI alone [23], improved by adding a contralateral HA [17]. Most studies found substantial bimodal benefits, ranging up to 30% on monosyllable word testing in quiet [17, 24–26]. In challenging listening conditions, such as when speech recognition is tested amongst background noise, the overall bimodal benefit tends to be even greater [24–26]. However, studies show that there is substantial variation in the extent to which CI patients benefit from wearing a contralateral HA [17, 21, 24]. Some subjects do not show improved speech intelligibility despite aidable residual hearing, or even perform worse with CIHA compared to CI alone [17, 24–26].

It is not yet understood which features of auditory perception explain individual differences in bimodal benefit. It seems intuitive that the effectiveness of bimodal aiding depends on the degree of residual hearing that can be stimulated acoustically with the HA. However, studies have reported mixed results on the relationship between bimodal benefit and audiometric thresholds, both aided and unaided [17, 21, 27–31]. Many other factors have been proposed, such as spectral resolution [27, 32], fundamental frequency processing [32] and monaural speech intelligiblity scores [33, 34], but uncertainty still exists.

**Bimodal loudness.** Currently, standardised methods for bimodal fitting are present, but these are brand-specific and not accepted by all manufacturers. Surveys show that the majority of clinicians do not apply specific HA fitting in case of bimodal wearing [35, 36]. This might be explained by the high variability in performance between subjects and the lack of predictive parameters for bimodal benefit. Specific adjustments to the HA however, and in some cases perhaps also to the CI, can be expected to benefit integration between CI and HA. As reviewed by Vroegop et al. [37], bimodal fitting strategies that have been proposed mostly focus on either alterations of frequency response, frequency transposition, frequency compression, or loudness. Studies on loudness mostly concentrate on implementing a loudness balancing strategy between CI and HA, thereby minimizing differences between the two sides [38, 39]. There is however no consensus yet on whether loudness strategies could be a valuable approach for bimodal fitting. Also, different procedures exist.

From a semantic standpoint, in this study we describe two types of information involved in bimodal hearing when evaluating loudness. We refer to equivalent loudness for those sounds that are perceived equally loud by CI and HA, and to differential loudness for sounds that are perceived differently in terms of loudness. Most studies on bimodal loudness strategies focused on equalization of loudness. In general, one can assume that the processing of speech (and other auditory information) is best if loudness in both ears is equalized across frequencies as much as possible. If, however, adequate processing in one ear is severely restricted to some frequencies or integration of information across ears is hampered by other reasons, it may be beneficial to avoid loudness equalization and focus on loudness differentiation, where fitting is adjusted in such a way that CI and HA optimally supplement each other by making use of the strengths of both devices. It is currently not yet understood which mechanism optimizes the benefits of bimodal hearing in these patients.

## Current study

In this study, a loudness scaling procedure was used to measure loudness growth with CI and HA in a group of adult bimodal subjects. Using a newly developed loudness function, individual loudness growth curves were constructed for each modality. These were integrated across the frequency spectrum and interpolated in a three-dimensional space using spline functions. The resulting data were then visualized in a colored graph: the loudness audiogram. This study had multiple aims. 1) To explore loudness growth differences between modalities (CI vs HA) in a group of bimodal subjects. 2) To relate loudness growth to speech recognition outcomes, such as speech intelligibility in quiet and noise, listening ease, and sound quality. These will be assessed both within the same listening condition (e.g. loudness growth with CI and speech recognition with CI) and bimodally (taking into account different properties of loudness across CI and HA, e.g. the extent to which HA was louder than CI and bimodal benefit). 3) Evaluate the loudness audiogram and analyze the additional benefit of measuring the full course of loudness growth compared to dynamic range (MCL-threshold). 4) Evaluate if measuring time of loudness scaling can be reduced by using broadband instead of narrowband stimuli, thereby reducing the amount of tests within the procedure.

## Methods

### Ethics

This study has been approved by the ethics committee of the Maastricht University Medical Center (MUMC+) under registration number NL42011.068.13 and has been registered in the Dutch National Trial Register (NTR3932). Subjects provided written informed consent before participation and were compensated for their travelling costs.

### Subjects

Fifteen adult bimodal listeners (8 male, 7 female, mean age: 62 years (standard deviation 12.5 years)) participated in this study. All subjects were post-lingually deafened, fluent speakers of the Dutch language and had at least one year of experience with a CI of the brand Advanced Bionics (Valencia, US). CI fitting was performed according to clinical routine with real life adjustments based on behavioral M and T levels. All participants declared consistent use of a contralateral HA with different brands worn between subjects. During testing, they used their own hearing devices, at typical daily use settings and manipulations during the course of testing were not allowed. Hearing aids were fitted separately from CI, as part of clinical routine, either within the clinic or at a local hearing aid professional. No systematic bimodal fitting protocol was applied, as no generally accepted bimodal fitting methods exist. Both the CI speech processor and the HA were checked to ensure they were working correctly. Unaided audiometry showed considerable residual hearing in the contralateral ear with thresholds up to 1000 Hz on average (Fig 1, extracted from Devocht et al. [17]). Mean pure-tone average (PTA) across 250, 500, and 1000 Hz on the HA-side was found to be 81.6 dB HL (SD: 18.3 dB) in the unaided and 36.0 dB HL (SD: 7.4 dB) in the aided situation. For further details on the participants' characteristics and hearing situation, see Devocht et al. [17].

### Procedures

All measurements were performed in a sound attenuated booth with subjects using their own hearing devices at daily use settings with no manipulations allowed during testing. When testing monaurally, the contralateral device was turned off and left in situ. The main outcome of

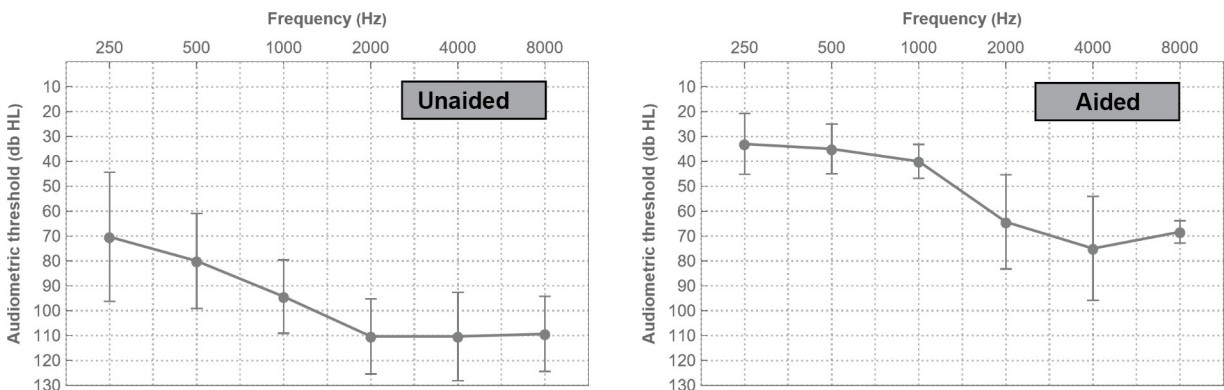

**Fig 1. Mean pure-tone air conduction thresholds in the hearing aid ear for the unaided and aided situation in free field.** Error bars indicate one standard deviation.

bimodal benefit was defined as the additional value of listening with CI and HA together compared to listening with CI alone. An overview of outcomes is shown in Table 1.

**Loudness and the loudness audiogram.** In this study, the Adaptive Categorical Loudness Scaling (ACALOS) procedure was used, which is a fast method and does not require training thereby making it applicable for clinical use [40]. For each input frequency, test results are usually presented as loudness curves relating stimulus intensity level (x-axis, dB HL) to loudness perception (y-axis, categorical units) across the subjects' dynamic range.

ACALOS was measured with CI and HA alone using the Oldenburg Measurement Applications (OMA) software (HörTech gGmbH, Oldenburg). Subjects were presented with two types of stimuli: narrowband (NB) noise (with 1/3-octave bandwidth at 250, 500, 1000, 2000 Hz) and broadband (BB) noise. Results measured with NB and BB noise will be cited as ACALOS$_{NB}$ and ACALOS$_{BB}$ respectively. The BB signal consisted of a modification of the

**Table 1. Overview of study outcomes.**

| | | | MODALITIES | | | | BIMODAL BENEFIT |
|---|---|---|---|---|---|---|---|
| OUTCOME MEASURE | PROCEDURE | CONDITIONS | CI | HA | CIHA | CONDITIONS | CIHA-CI |
| **Loudness scaling** | | | | | | | |
| Narrowband (NB) | ACALOS (OMA) | 250, 500, 1000, 2000 Hz | X | X | | | |
| Broadband (BB) | ACALOS (OMA) | IFFM | X | X | | | |
| **Speech outcomes** | | | | | | | |
| Speech in quiet | CNC (% phonemes correct) | max (over 55-65-75dB SPL) | X | X | X | Normalized benefit | X |
| Speech in noise | Dutch Matrix Sentence Test (iSRT) | S0N0 | X | | X | Summation (S0N0), Head shadow (S0NCI) | X |
| Listening ease | Categorical rating scale (OMA) | +9 SRT | X | | X | +9 SRT | X |
| Speech quality | Questionnaire | Full, pleasant, tinny | X | X | X | Full, pleasant, tinny | X |
| **Audiometry** | | | | | | | |
| Audiometric thresholds | Free field warble tones | 250, 500, 1000, 2000 Hz | X | X | | | |

CI = Cochlear implant, HA = Hearing aid, CIHA = bimodal, CNC = consonant nucleus consonant, OMA = Oldenburg Measurement Applications software, S0N0 = speech and noise presented from front (0˚), S0NCI = speech presented from front (0˚), noise presented from CI side, iSRT = inverted speech reception threshold, +9 SRT = presented at a speech-to-noise ratio of 9 dB above speech reception threshold (measured for speech in noise in same listening condition).

International Female Fluctuating Masker (IFFM) [41, 42]. The IFFM consists of a multilingual voice signal that has the spectral and temporal characteristics of a single speaker but is non-intelligible as a whole. In the modified version, the fundamental frequency of the IFFM signal was decreased to male standards (127 Hz) to allow for extra information in the lower frequency range of aidable residual hearing in bimodal users [43]. Stimuli were presented at different intensity levels (range 0–95 with dB HL for NB and dB SPL for BB) from a loudspeaker positioned 1m in the front of the seated subject at ear level. Subjects were instructed to rate loudness perception on a touch screen with the 11-point ACALOS scale ranging from inaudible to too loud. Each loudness category was mapped to categorical units (CU) from 0 to 50, which were not visible to the subject. The adaptive ACALOS procedure was used, which adjusts stimulus intensity to the subjects' individual auditory range and presents levels in a randomized order [40]. It consists of two phases, where in the first phase the dynamic range is estimated. To reach the upper limit (response 'too loud'), stimulus level is increased in steps of 10 dB until 90 dB HL, then in 5 dB steps, until the desired response is reached or maximum stimulus level is presented. To find threshold level (response 'not heard'), stimulus level is decreased with steps of 15 dB until it was inaudible, and then increased with 5 dB steps until it was audible again [40]. In the second phase, stimulus levels within the dynamic range are estimated by linear interpolation and presented in randomized order [40].

Due to the non-linear characteristics of loudness growth curves, a simple linear model does not provide an optimal fit. Different fits are available but the applicability of each function depends on the measurement condition (free field or headphones), listening condition (aided or unaided) and subject characteristics (normal hearing or hearing impaired) [44–46]. Evaluation of the loudness functions however, has mainly been done with unaided hearing and using headphones. When loudness scaling is being performed in free field and using hearing aids, some well-known factors have to be accounted for. For example, stimulation range is smaller in free field (0–95 db SPL) versus headphones (0–120 db SPL). Both hearing aids and cochlear implants can alter the shape of the loudness curve due to individual settings (i.e. maximum comfortable levels), compression rules and output limiting functionalities [46, 47]. Therefore, in this study a newly developed loudness function was introduced. It aims for higher accuracy and less bias compared to current fitting functions when loudness scaling is performed in free field using hearing aids or cochlear implants. A detailed step-by-step description of the fit is shown in Table 2. The loudness function was programmed in Mathematica 12.3 (Wolfram Research, Champaign, USA).

Loudness growth curves were constructed for NB and BB (Table 2, B3). The curves based on $ACALOS_{NB}$ were then integrated (Table 2, C1) and mapped three-dimensionally using a spline function (Table 2, C2). The resulting set of coordinates was visualized as a loudness audiogram, each containing properties of frequency (Hz), stimus level (dB HL) and loudness perception (CU)(Table 2, C3). Besides loudness audiograms for each CI and HA measurement, also a third visualization was made in which the differences between per device are shown.

For each outcome of loudness growth, measures of area (2D, $ACALOS_{BB}$) or volume (3D, $ACALOS_{NB}$) were calculated to account for the complete course of loudness build-up. For BB, an Area Under the Curve (AUC) was calculated. For the loudness audiograms (NB), an Area Under the Surface (AUS) was calculated by determining the summed loudness perception (CU)(Table 2, D1).

Since multiple factors might influence bimodal benefit, different properties of loudness information across CI and HA were considered for analysis. Fig 2, which is a hypothetical example of loudness growth curves measured separately with CI and HA (in 2D for simplicity purposes), illustrates these variables. *CI* and *HA* are defined as the areas below CI and HA

**Table 2. Algorithm for extending the audiogram with loudness growth.**

A. Perform loudness scaling using the standard ACALOS procedure (see 2.3.1. Loudness and the loudness audiogram)

B. Curve fitting

| | Step | Pseudocode | Result |
|---|---|---|---|
| B1 | Determine the threshold of the loudness growth curve | | |
| B1.1 | Take the x (stimulus intensity level (db SPL)) values of all cases where the y (loudness perception (CU)) value is 0. If no 0 is present, a 0 is added to start. | *X values with (Y value = 0)* | *Table 2 B1p1.tif* |
| B1.2 | Take the x values of all cases where the y value is not 0 (range 1–50) | *X values with (Y value ≠ 0)* | *Table 2 B1p2.tif* |
| B1.3 | Determine the cut-off | *Mean of the points of the complement of B1.1 with B1.2* | *Table 2 B1p3.tif* |
| B1.4 | Remove data before cut-off | *Remove points with X < B1.3* | |
| B1.5 | Prepend with zeroes starting at (0, 0), ending a distance of 20 before the cut-off point | *Add 6 points with Y = 0, from (0,0) until B1.3$_{X-20}$* | *Table 2 B1p5.tif* |
| B2 | Take the moving median of values. Subsequently apply a moving average to smooth and create an interpolation function over the resulting points. | *Interpolate (order 1) the MovingAverage of a MovingMedian of B1.5* | |
| B3 | Determine the values over the full 0–95 range, clipped at the maximum measured Y value. | *Clip B1.5$_Y$ (0-Max[B1.2$_y$]) for X 0 to 95 (stepsize 1)* | |
| | Plot loudness growth function for a given frequency band. | | *Table 2 B3. tif* |
| | Legend: B1.1 (red dots), B1.2 (blue dots), B1.3 (brown line), B1.5 (black dots), estimated threshold defined as 5 CU (T5CU, red line), loudness fit (purple line). | | |

C. Integration & Visualisation

| | Step | Pseudocode | Result |
|---|---|---|---|
| C1 | Integration of loudness growth curves per frequency band | | *Table 2 C1. tif* |
| C2 | Integration using Spline interpolation | | *Table 2 C2. tif* |
| C3 | The loudness audiogram: colored display with x-axis = frequency (Hz), y-axis = stimulus intensity level (dB HL). Loudness perception (CU) is color-coded. Aided audiometric thresholds (previously measured in in clinical routine) shown with dashed black line and estimated ACALOS$_{NB}$ thresholds (T5CU) in dashed gray line. | | *Table 2 C3. tif* |

D. Area under the surface calculation

| | Step | Pseudocode | Result |
|---|---|---|---|
| D1 | Calculation of loudness growth in the loudness audiogram by taking the loudness perception for each coordinate (dB HL, Hz). | | 46647 |

growth curves, respectively, when evaluated separately. *CI+* represents the area where CI dominates HA in terms of loudness, whereas *HA+* represents the area where HA dominates CI. The area where loudness is induced by both devices is captured by *Overlap*. By taking the sum of *CI+*, *HA+* and *Overlap* the *Total CI + HA* is calculated, representing the imputed overall loudness for both devices and total available information in terms of loudness. As introduced in the introduction, Fig 1 shows that equivalent loudness is captured by the property *Overlap*, and differential loudness is captured by *CI+* and *HA+*. Theoretically, in case of perfectly matched loudness, Overlap would be 100%. Contrarily, if low frequencies were only audible by the HA and high frequencies only by the CI, then Overlap would be 0% and contributions of HA+ and CI+ would be substantial.

**Performance within the same listening condition and bimodal benefit.** Relationships between loudness and speech outcomes were studied for both performance within the same listening condition and bimodal benefit. Bimodal benefit was defined as the difference in speech outcome results between CIHA and CI alone. Only those outcomes (or subdomains) that showed significant bimodal benefit in Devocht et al. [17] were included for analysis in this study. These included: CNC word recognition, sentence intelligibility in noise (inverted SRT (iSRT), in different spatial conditions), ease of listening (inverse of listening effort) at SRT +9 dB SNR, and sound quality for the ratings *Full*, *Pleasant* (inverse of *Unpleasant*) and *Tinny*.

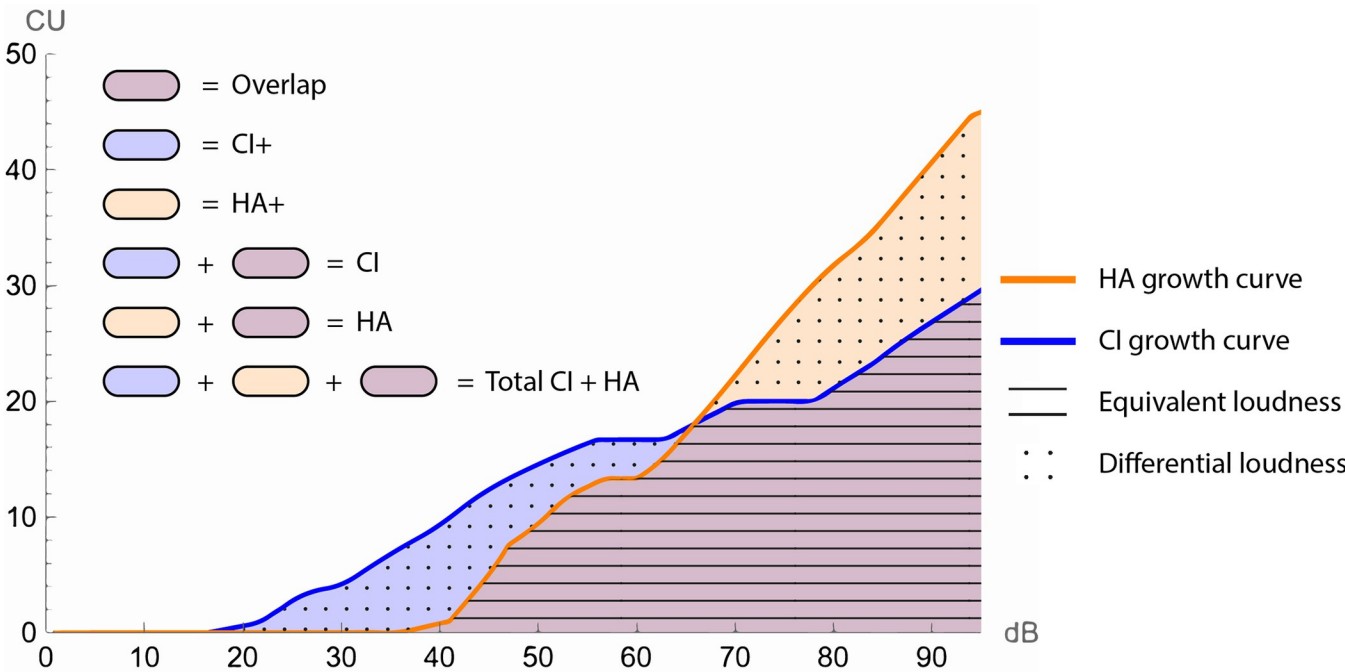

**Fig 2. Graph illustrating unilaterally measured loudness growth curves of CI and HA showing the different properties of loudness information that were evaluated in this study.**

Testing procedures and results are previously described extensively in Devocht et al. [17]. In short, word intelligibility in quiet was retrieved from the last clinical routine measurement by recording the maximum score on a Dutch monosyllabic consonant-nucleus-consonant (CNC) over the levels 55, 65, and 75 dB SPL in quiet from the frontal direction. Sentence intelligibility in noise (Speech Reception Threshold (SRT)) was measured using the Dutch matrix test, with speech and noise (a stationary noise with power spectrum equivalent to speech input) in multiple spatial conditions. Listening effort was subsequently measured in OMA with subjects rating effort on a categorical response scale at multiple speech to noise ratios. Sound quality was evaluated with a translated questionnaire [17] of quantifiable sound quality attributes originally described by Boretzki [48]. All outcomes were measured using CI and CIHA. Sound quality and CNC were also measured with HA alone. Outcomes were inverted where necessary such that a more positive value reflected a more favorable outcome throughout. In order to deal with ceiling effects, CNC word recognition scores were converted into normalized bimodal benefit, as originally proposed by Zhang et al. [27]. Binaural effects were calculated by subtracting SRT outcomes with CIHA and CI measured in the S0NCI condition for bimodal head shadow and in the S0N0 condition for binaural summation. Speech recognition in noise (dB SNR at SRT), as measured with CI and CIHA, was also inverted so a higher score reflected a higher speech recognition ability.

## Data analysis

All statistical analyses were performed with Mathematica 12.3. Given the small sample size of this study, analysis was limited to descriptive statistics and an explorative correlation analysis using non-parametric Spearman rank coefficients without any predefined corrections. Correlations were calculated between the loudness AUS ($ACALOS_{NB}$ and $ACALOS_{BB}$) and speech outcomes (represented as performance within the same listening condition and bimodal

benefit). Dynamic range was estimated from $ACALOS_{NB}$ data by calculating the difference between stimulus intensity levels (dB HL) at threshold (T5CU) and at the first position where maximum CU was reached. The average dynamic range across frequencies for both HA and CI was recorded as well as a difference score between the two devices (HA minus CI). Relationships were examined between dynamic range and bimodal benefit to explore the advantage of measuring loudness growth instead of only using threshold and maximum level. Furthermore, correlations were calculated between aided audiometric thresholds and both loudness AUS as well as T5CU. To explore relations between loudness measured with $ACALOS_{NB}$ and $ACALOS_{BB}$, results of both methods were compared.

## Results

### Loudness and the loudness audiogram

Fitting the ACALOS data, generating the loudness growth curves and constructing the loudness audiograms involved several steps (see Table 2). S1 Fig shows all individual loudness growth curves measured with $ACALOS_{NB}$ and $ACALOS_{BB}$ (original loudness data in supporting dataset S1 Dataset). An example of the newly developed loudness function in comparison to existing model functions is shown in S2 Fig. Deviations between ACALOS data points and individual loudness functions were assessed for each curve by calculating the Root Mean Square Error (RMSE). Mean RMSE and 95% confidence interval was 2.72 (2.57–2.87) for NB and 2.68 (2.30–3.07) for BB. Fig 3 shows the resulting loudness audiograms for all subjects. AUS's and AUC's can be found in S1 and S2 Tables.

### Qualitative analysis on bimodal loudness

The loudness audiograms show a wide variety of combined information per modality and across patients. Median ratios of loudness growth across different modalities show that 70% (min 26%, max 85%) of total information included Overlap (S1 Table). Remaining information was distributed across CI+ (median 13%, min 0%, max 45%) and HA+ (median 8%, min 0%, max 74%). Taking into account the small study sample size, trends in data were observed with visual examination of the loudness audiograms. Here, differences in loudness growth between devices across frequencies and loudness levels were observed. Relationships with speech performance in quiet and noise, as published in Devocht et al. [17] and shown in S3 Table, are also discussed.

In some patients, loudness was mostly dominated by the CI. This is most striking for subject B06 (CI+ 45%, HA+ 0%), which can be explained by significantly better thresholds for CI than HA across all measured frequencies. This subject showed relatively high speech recognition with CI, but only a minor additional benefit of wearing a contralateral HA. Loudness audiograms of subjects B03 (CI+ 20%, HA+ 2%) and B37 (CI+ 27%, HA+ 2%) show slightly better thresholds for CI compared to HA, especially in the high frequencies. Also, dynamic range with CI was higher. Surprisingly, subject B03 still had no monaural speech recognition with CI, therefore precluding evaluation of bimodal benefit.

In other subjects, loudness was dominated by the HA thereby providing differential cues to the CI. For example, subject B45 (CI+ 0%, HA+ 74%) showed both superior thresholds and more loudness with HA compared to CI. This is reflected in only minor speech recognition with CI and relatively high bimodal benefit when assessed in quiet. Performance in noise with CI, and therefore also bimodal benefit, could not be assessed. In subjects B08 (CI+ 2%, HA+ 61%) and B22 (CI+ 6%, HA+ 40%) HA thresholds were better at all frequencies except 2000 Hz. Also, higher levels of maximum loudness were perceived with HA. Both subjects reached

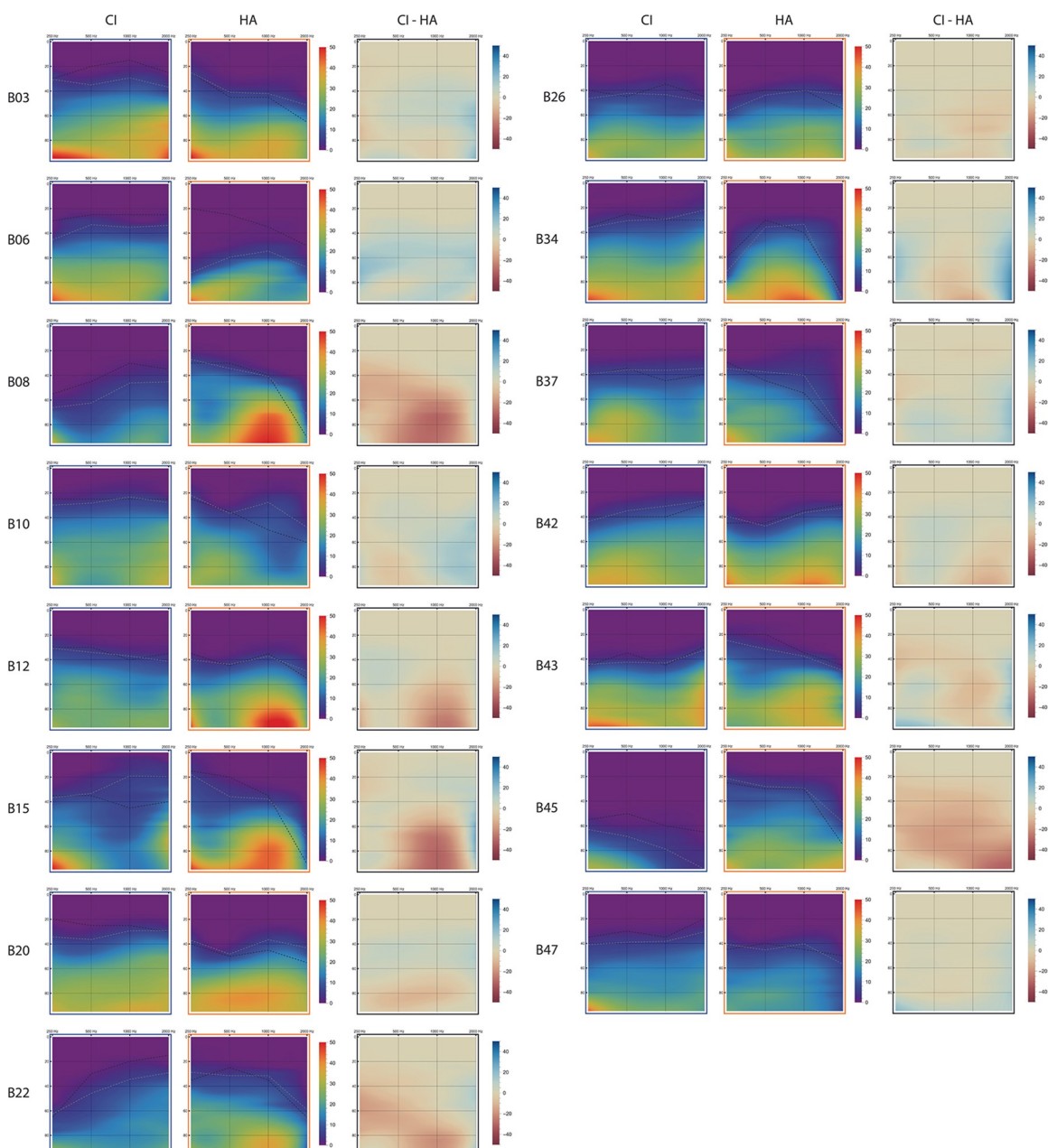

**Fig 3. Loudness audiograms for all conditions (CI (blue frame), HA (orange frame) and CI minus HA (black frame).** Aided audiometric thresholds are projected as dashed black lines and estimated ACALOS thresholds (T5CU) as gray lines.

relatively high bimodal benefit for summation and head shadow effect. Bimodal benefit in quiet was high for subject B22, but close to average for subject B08.

For other patients, dominance of either CI or HA varied across frequencies. Subject B34 (CI+ 28%, HA+ 5%) for example, had superior thresholds with CI at low and high frequencies, while at mid frequencies both devices showed similar thresholds but loudness growth appeared more pronounced with HA. Monaural speech recognition with CI and HA was close to group median for this subject, and bimodal benefit was lower. In subjects B10 (CI+ 28%, HA+ 4%) and B12 (CI+ 8%, HA+ 22%), thresholds for CI and HA were similar. Loudness in the high

frequencies however, was dominated by the CI in subject B10, and by the HA in subject B12. Subject B10 had a bimodal benefit which was above group median for summation, but slightly lower for head shadow and speech in quiet. Subject B12 scored below median for bimodal benefit in quiet, with bimodal benefit not assessed. In subject B20 (CI+ 13%, HA+ 8%), dominance of CI and HA did not differ across frequencies, but did at different input intensity levels. Although thresholds were better with CI, maximum loudness was not reached with CI but only with HA. This subject only reached minor bimodal benefit.

## Exploratory correlation analysis

**Loudness and performance within the same listening condition.** As shown in Table 3, a significant correlation was found between speech recognition in noise and loudness (BB) for CI where a higher AUS corresponded with improved speech recognition ($r_s$ = 0.67, p = 0.02). Also, speech was perceived as less *tinny* with CIHA when there was more loudness with CI +HA ($r_s$ = -0.54, p = 0.04).

**Loudness and bimodal benefit.** *Speech recognition*. No significant correlations between loudness AUS and bimodal benefit in speech in quiet were found (Table 4). However, for speech recognition in noise, Spearman correlation revealed significant relationships with certain properties of loudness. Less loudness with CI (CI, $r_s$ = -0.67, p = 0.02 / CI+, $r_s$ = -0.82, p = 0.00) and more loudness with HA (HA+, $r_s$ = 0.64, p = 0.04) was significantly correlated with a higher head shadow effect, calculated as SRT CI minus CIHA measured in the S0NCI condition. Binaural summation was most effective when loudness (NB) was low in CI+ ($r_s$ = -0.79, p = 0.00) and high in HA+ ($r_s$ = 0.69, p = 0.02). For BB, the same trend was found for low loudness in CI ($r_s$ = -0.61, p = 0.05). That is, more binaural summation was related to differential loudness information of the hearing aid in loudness growth. Dynamic range (individual results in S4 Table), calculated as the difference between stimulus levels at threshold (T5CU) and maximum reached CU, showed similar relationships with bimodal benefit as

**Table 3. Correlations between loudness (AUS for $ACALOS_{NB}$ and AUC for $ACALOS_{BB}$) and performance within the same listening condition (speech recognition, listening ease and sound quality) within the same modality (CI, HA and Total CI+HA).**

| | | Loudness | | | | | |
| | | Narrowband (NB) | | | Broadband (BB) | | |
| | | CI | HA | Total CI+HA | CI | HA | Total CI+HA |
|---|---|---|---|---|---|---|---|
| Speech in quiet (CNC % correct, N = 14) | Rho | 0.10 | 0.46 | -0.17 | -0.09 | 0.45 | -0.28 |
| | P | 0.72 | 0.08 | 0.54 | 0.75 | 0.09 | 0.32 |
| Speech in noise (iSRT, dB SNR, N = 11) | Rho | 0.67 | NM | 0.04 | 0.31 | NM | -0.08 |
| | P | 0.02* | NM | 0.90 | 0.36 | NM | 0.79 |
| Listening ease at +9 SRT (rating scale, N = 14) | Rho | -0.29 | NM | 0.02 | 0.06 | NM | 0.38 |
| | P | 0.29 | NM | 0.95 | 0.83 | NM | 0.17 |
| Full (rating scale, N = 14) | Rho | 0.29 | -0.06 | -0.03 | 0.40 | 0.02 | 0.27 |
| | P | 0.29 | 0.82 | 0.91 | 0.14 | 0.94 | 0.32 |
| Pleasant (rating scale, N = 14) | Rho | 0.40 | 0.15 | 0.43 | 0.12 | 0.02 | 0.09 |
| | P | 0.14 | 0.59 | 0.11 | 0.68 | 0.93 | 0.75 |
| Tinny (rating scale, N = 14) | Rho | -0.20 | -0.46 | -0.54 | -0.21 | 0.03 | -0.31 |
| | P | 0.46 | 0.08 | 0.04* | 0.44 | 0.92 | 0.27 |

Speech in noise with CI was limited to 11 subjects due to 4 subjects not being able to perform matrix in noise testing. Grey shade levels are applied according to Spearman's rho classifications as published by Dancey and Reidy [49]. NM = Not Measured.

*p<0.05.

**Table 4. Correlations between loudness growth (AUS for $ACALOS_{NB}$ and AUC for $ACALOS_{BB}$) and bimodal benefit on various outcomes.**

| | | Loudness | | | | | | | | | | | |
| --- | --- | --- | --- | --- | --- | --- | --- | --- | --- | --- | --- | --- | --- |
| | | Narrowband (NB) | | | | | | Broadband (BB) | | | | | |
| | | CI | HA | Overlap | CI+ | HA+ | Total CI+HA | CI | HA | Overlap | CI+ | HA+ | Total CI+HA |
| Speech in quiet (CNC % correct, N = 14) | | | | | | | | | | | | | |
| Normalized benefit | Rho | -0.06 | 0.15 | 0.03 | -0.40 | 0.32 | 0.00 | -0.34 | 0.45 | 0.15 | -0.31 | 0.39 | 0.01 |
| | P | 0.83 | 0.58 | 0.91 | 0.14 | 0.25 | 1.00 | 0.21 | 0.10 | 0.59 | 0.27 | 0.16 | 0.96 |
| Speech in noise (iSRT, dB SNR, N = 11) | | | | | | | | | | | | | |
| Head shadow | Rho | -0.67 | 0.47 | -0.35 | -0.82 | 0.64 | -0.18 | -0.71 | -0.02 | -0.25 | -0.28 | 0.46 | -0.36 |
| | P | 0.02* | 0.14 | 0.28 | 0.00* | 0.04* | 0.59 | 0.01* | 0.96 | 0.45 | 0.40 | 0.16 | 0.27 |
| Summation | Rho | -0.59 | 0.59 | -0.24 | -0.79 | 0.69 | 0.02 | -0.61 | 0.16 | -0.02 | -0.52 | 0.36 | -0.48 |
| | P | 0.06 | 0.06 | 0.48 | 0.00* | 0.02* | 0.96 | 0.05* | 0.65 | 0.96 | 0.10 | 0.28 | 0.14 |
| Listening effort (rating scale, N = 14) | | | | | | | | | | | | | |
| Listening ease at +9 SRT | Rho | -0.54 | -0.28 | -0.49 | -0.21 | -0.05 | -0.53 | -0.34 | -0.26 | -0.48 | 0.02 | 0.22 | -0.31 |
| | P | 0.04* | 0.32 | 0.06 | 0.45 | 0.85 | 0.04* | 0.21 | 0.35 | 0.07 | 0.94 | 0.43 | 0.26 |
| Speech quality (rating scale, N = 14) | | | | | | | | | | | | | |
| Full | Rho | -0.31 | 0.62 | -0.07 | -0.54 | 0.69 | 0.20 | -0.34 | 0.08 | -0.12 | -0.40 | 0.42 | -0.02 |
| | P | 0.26 | 0.01* | 0.81 | 0.04* | 0.00* | 0.48 | 0.21 | 0.79 | 0.67 | 0.14 | 0.12 | 0.95 |
| Pleasant | Rho | -0.17 | 0.48 | -0.04 | -0.34 | 0.31 | 0.24 | -0.12 | -0.05 | -0.48 | -0.06 | 0.13 | -0.10 |
| | P | 0.53 | 0.07 | 0.89 | 0.21 | 0.27 | 0.39 | 0.66 | 0.87 | 0.07 | 0.84 | 0.65 | 0.74 |
| Tinny | Rho | 0.24 | -0.06 | 0.31 | 0.15 | -0.13 | -0.01 | 0.49 | -0.05 | 0.35 | 0.13 | -0.28 | 0.46 |
| | P | 0.39 | 0.84 | 0.25 | 0.60 | 0.64 | 0.96 | 0.07 | 0.86 | 0.19 | 0.64 | 0.31 | 0.08 |

Speech in noise with CI was limited to 11 subjects due to 4 subjects not being able to perform matrix in noise testing. Grey shade levels are applied according to Spearman's rho classifications as published by Dancey and Reidy [49].

*p<0.05.

AUS, but some differences existed (Table 5). For example, dynamic range for the HA was significantly correlated with bimodal benefit in speech in quiet ($r_s$ = 0.58, p = 0.02) and binaural summation ($r_s$ = 0,71, p = 0.02) while this was not the case with AUS.

*Listening ease*. Significant correlations were found between listening ease and loudness (NB) where more loudness with CI ($r_s$ = -0.54, p = 0.04) and Total CI+HA ($r_s$ = -0.53, p = 0.04) corresponded with reduced bimodal benefit. Thus, the additional benefit of wearing a HA in terms of listening ease was lower when loudness with CI and CIHA was high. No significant relationship between dynamic range and listening ease was found.

*Speech quality*. Speech was perceived as significantly more *full* (with CIHA compared to CI) when loudness (NB) was high in HA ($r_s$ = 0.62, p = 0.01), HA+ ($r_s$ = 0.69, p = 0.00) and low in CI+ ($r_s$ = -0.54, p = 0.04). In other words, a dominance of the HA over the CI in terms of perceived loudness appeared to be related to a fuller speech experience. This finding was also found when comparing with the difference score between dynamic range with HA and CI ($r_s$ = 0.56, p = 0.03).

## Loudness and audiometric thresholds

A significant correlation was found between aided audiometric thresholds with CI averaged across 250-500-1000 Hz and loudness (NB) with CI ($r_s$ = 0.66, p = 0.01) (Table 6). Thus, more loudness AUS corresponded with better audiometric thresholds. For HA, no significant relationship was found between both measures. Correlations were also calculated between audiometric thresholds and $ACALOS_{NB}$ thresholds (T5CU) (Table 7). Here, 250 and 500 Hz correlated for both CI and HA and 2000 Hz only for HA.

**Table 5. Correlations between dynamic range (difference between stimulus intensity levels (dB HL) at threshold (T5CU) and at the first position where maximum CU was reached, averaged across frequencies) and bimodal benefit on various outcomes.**

| | | Dynamic range | | |
|---|---|---|---|---|
| | | CI | HA | Difference (HA—CI) |
| Speech in quiet (CNC % correct, N = 14) | | | | |
| Normalized benefit | Rho | -0.08 | 0.58 | 0.51 |
| | P | 0.79 | 0.02* | 0.05* |
| Speech in noise (iSRT, dB SNR, N = 11) | | | | |
| Head shadow | Rho | -0.48 | 0.49 | 0.65 |
| | P | 0.13 | 0.13 | 0.03* |
| Summation | Rho | -0.39 | 0.71 | 0.83 |
| | P | 0.24 | 0.02* | 0.00* |
| Listening effort (rating scale, N = 14) | | | | |
| Listening ease at +9 SRT | Rho | -0.43 | -0.19 | -0.03 |
| | P | 0.11 | 0.49 | 0.91 |
| Speech quality (rating scale, N = 14) | | | | |
| Full | Rho | -0.26 | 0.47 | 0.56 |
| | P | 0.35 | 0.07 | 0.03 |
| Pleasant | Rho | -0.39 | 0.07 | 0.27 |
| | P | 0.15 | 0.79 | 0.34 |
| Tinny | Rho | 0.23 | -0.15 | -0.18 |
| | P | 0.40 | 0.60 | 0.52 |

Speech in noise with CI was limited to 11 subjects due to 4 subjects not being able to perform matrix in noise testing. Grey shade levels are applied according to Spearman's rho classifications as published by Dancey and Reidy [49].

*p<0.05.

### Loudness narrowband versus broadband

Loudness measured with ACALOS$_{NB}$ and ACALOS$_{BB}$ were significantly correlated in CI ($r_s$ = 0.67, p = 0.01), Overlap ($r_s$ = 0.54, p = 0.04) and HA+ ($r_s$ = 0.58, p = 0.02) (Table 8). That is, measuring loudness with either narrowband or broadband noise produced similar results for those properties, but not for HA, CI+ and Total CI+HA.

## Discussion

In this article, we presented the concept of the loudness audiogram. By extending the conventional audiogram with loudness growth, a visual instrument was created that not only displays

**Table 6. Correlations between loudness (AUS for ACALOS$_{NB}$ and AUC for ACALOS$_{BB}$) and aided audiometric thresholds within the same modality (CI and HA).**

| | | Loudness | | | |
|---|---|---|---|---|---|
| | | Narrowband (NB) | | Broadband (BB) | |
| Aided thresholds | | CI | HA | CI | HA |
| Low (250-500-1000 Hz) | Rho | 0.66 | 0.25 | 0.38 | -0.18 |
| | P | 0.01* | 0.38 | 0.16 | 0.52 |
| High (500-1000-2000 Hz) | Rho | 0.47 | -0.10 | 0.43 | -0.31 |
| | P | 0.08 | 0.73 | 0.11 | 0.26 |

*p<0.05.

**Table 7. Correlations between estimated $ACALOS_{NB}$ threshold values (T5CU) and aided audiometric thresholds within the same modality (CI and HA).**

| | | ACALOS$_{NB}$ thresholds | |
|---|---|---|---|
| | | CI | HA |
| Aided audiometric thresholds with corresponding device | | | |
| 250 Hz | Rho | 0.83 | 0.64 |
| | P | 0.00* | 0.01* |
| 500 Hz | Rho | 0.66 | 0.57 |
| | P | 0.01* | 0.03* |
| 1000 Hz | Rho | 0.39 | 0.20 |
| | P | 0.15 | 0.47 |
| 2000 Hz | Rho | 0.41 | 0.70 |
| | P | 0.13 | 0.00* |

*p<0.05.

threshold values, but also presents the full span of the dynamic range across a frequency spectrum. The relationship between loudness growth and speech outcomes was evaluated.

## Bimodal loudness

A qualitative and quantitative analysis was performed to identify the predictive value of loudness growth measurements on speech outcomes and study the combination of loudness between electric and acoustic hearing in bimodal patients. Currently, it is unclear how information from CI and HA can optimally be combined to achieve best hearing performance. That is, whether fitting strategies should focus on optimizing equalization and matching of loudness, or, in contrast, on optimizing differentiation. In the framework of this study, both mechanisms were identified within the domain of loudness by using loudness audiograms of bimodal patients. In terms of the different properties posed in this study, equalization of loudness would entail high AUS for Overlap and low AUS for CI+ and HA+. In contrast, differentiation of loudness could be characterized as smaller AUS for Overlap, but more AUS for CI + and HA+.

Visual examination of loudness data showed different patterns. Arbitrarily, there seemed three types of bimodal subjects; for whom the CI was dominant in terms of loudness, for whom HA was dominant, and for whom dominance of either CI and HA depended on frequency and loudness level. Although limited by a small sample size, correlation analysis showed that Overlap was not significanty related with any speech outcome while CI+ and HA + showed many relationships with bimodal benefit (Table 4). This was most prominent for speech in noise testing, where high differentiation (CI+ and HA+) was highly predictive for head shadow and summation effect but there was no relationship with Overlap. Also, the

**Table 8. Correlations between loudness growth measured with $ACALOS_{NB}$ and $ACALOS_{BB}$ within the same property (CI, HA, Overlap, CI+, HA+ and Total CI +HA).**

| | | Loudness (NB) | | | | | |
|---|---|---|---|---|---|---|---|
| | | CI | HA | Overlap | CI+ | HA+ | Total CI+HA |
| Loudness (BB) | Rho | 0.67 | 0.29 | 0.54 | 0.45 | 0.58 | 0.41 |
| | P | 0.01* | 0.30 | 0.04* | 0.09 | 0.02* | 0.13 |

*p<0.05.

speech quality fullness significantly correlated with differentiation (HA+/CI+) but not with Overlap or HA only. For listening effort, there were no significant correlations, although Overlap had better predictive value than CI+ and HA+.

**Clinical translation.** In general, results from our patient set suggest that loudness growth differentiation induced higher bimodal benefit than loudness equalization. From a clinical perspective, this would imply that bimodal fitting should not solely focus on balancing CI and HA in terms of loudness, but also on optimally utilizing the strengths of both devices across the available frequency spectrum. It should be noted however, that there are no recommendations available on when and how to adjust gain accordingly. Possibly, the optimal bimodal fitting might provide both equal and differential contributions of CI and HA depending on thresholds an frequency. However, the sample size in this study is too small to draw clinical conclusions. Likely, the effects of differentiation versus equalization also depend on the available amount of residual hearing. Candidacy criteria for cochlear implantation have expanded over the years [50], leading to increasing numbers of CI patients with lesser degrees of contralateral hearing loss. For these subjects, the availability of larger acoustic bandwidth with the HA might also affect interaction with the CI. For example, loudness equalization might be a more beneficial strategy to conserve Interaural Level Differences (ILD) than loudness differentiation. The magnitude of ILD however increases with the frequency of sound [51] thereby diminishing its relevance for bimodal fitting when only limited acoustic bandwidth is available.

## Loudness and performance within the same listening condition

Correlation analysis between loudness and speech outcomes measured within the same listening condition (with either CI, HA or CIHA) showed limited relationships. More loudness with CI correlated with higher speech recognition in noise with CI. Previously, unaided loudness recruitment was reported to be negatively correlated to speech recognition in noise [13, 14]. In these studies, loudness recruitment was defined as the early slope of the loudness curve while in the current study loudness growth was incorporated in full, making direct comparisons between studies difficult. Surprisingly, no relationships were found between loudness and both speech in quiet and listening ease within the same modality. For speech quality, subjects rated sound as less tinny when there was high loudness growth with CI+HA. Other speech outcomes measured with CIHA did not correlate with loudness growth. In part, this might be due to the fact that loudness was not actually measured with CIHA but derived by combining loudness measurements of CI and HA.

## Loudness and bimodal benefit

Relationships between loudness and bimodal benefit speech outcomes showed significant correlations. This was primarily the case for speech in noise, where bimodal benefit due to head shadow and binaural summation was more extensive when loudness with CI was limited and the HA could provide significant complementary loudness (HA+). A similar trend was found for speech in quiet, but without reaching significance. Bimodal benefit on ease of listening increased when there was less loudness available with CI and when the total information with CI+HA was high. Also, sounds were perceived as more *full* when more loudness with HA was observed. When loudness with CI was dominant (CI+), sound was recognized as less *full*, opposed to when loudness with the HA dominated (HA+). No trend was present for sound quality ratings *pleasant* and *tinny*. Interestingly, dynamic range showed similar relationships to bimodal benefit as loudness expressed in AUS. Specifically, the difference between dynamic range with HA and CI was significantly related to speech in noise outcomes. This suggests that dynamic range, as derived from ACALOS loudness data, on its own might already be a good

parameter to relate to bimodal benefit. It is undetermined if the same observation would be found if dynamic range was estimated with audiometric procedures since this was not tested. Also, the upper limit of the dynamic range was derived from loudness growth curves, taking into account plateau effects of loudness saturation by selecting the first stimulus intensity level at maximum loudness level. This is not a common procedure in conventional audiometry. Also, to evaluate how information from CI and HA are combined, the loudness audiogram provides a more detailed perspective, as illustrated by the different properties (CI/HA/Overlap/CI+/HA+/Total CI+HA) calculated in this study.

## Loudness and audiometric thresholds

In audiometry, thresholds are determined with pure tones (unaided) or warble tones (aided) while ACALOS uses one-third-octave band noises as stimuli (for NB condition). In ACALOS, hearing threshold is arbitrarily located around 5 CU (T5CU) which is the first category subjects can choose when stimulus recognition occurs (as 'very soft'). To evaluate differences in threshold assessment between standard audiometry and ACALOS, thresholds with both methods have been compared. Especially in the low frequencies (250 and 500 Hz), thresholds between audiometry and ACALOS were comparable. It should be noted however, that standard audiometry was performed earlier in time leaving the possibility for residual hearing to have further deteriorated before ACALOS measurements were performed. To evaluate whether standard audiometric thresholds and extended loudness growth were two distinct measures of auditory functioning, both variables were compared. Interestingly, only loudness (NB) with CI significantly correlated, suggesting loudness and audiometry provide discriminative input. In future research, audiometric thresholds should be measured in the same session as loudness. Likewise, a bigger study group should be used to further evaluate the distinctiveness of both measures.

## Loudness narrowband versus broadband

Measuring ACALOS with NB and BB produced different results. When comparing results with both methods, only significant correlations were reached on properties CI, Overlap and Total CI+HA. The biggest difference in loudness between NB and BB was noticeable for the HA, which in turn did not seem to be directly related to any specific frequency (S5 Table). Also, relationships with $ACALOS_{NB}$ and bimodal performance were more persistent than for $ACALOS_{BB}$. It can therefore be concluded that IFFM is not a complete replacement for NB measurements when assessing bimodal benefit with the ACALOS procedure.

## Loudness function

Previously, numerous functions to fit loudness data have been proposed, of which Brand [44] has tested many in terms of bias and accuracy. In both normal hearing and hearing impaired subjects, the model with the best results consisted of two linear sections connected at loudness value 25 CU which was smoothed with a Bezier interpolation between 15 and 35 CU [40]. More recently, Oetting et al. [45] developed an alternative loudness function. Both functions however are not designed for loudness measurements in free field and using hearing aids. Theelen et al. [46] proposed a new function for categorical loudness scaling in the electrical domain. In the current study, a new loudness function was introduced which aims to be more accurate when performed in free field under aided conditions than conventional fits. Future research should validate this function in a large group of subjects, by comparing it to current available fits in terms of goodness-of-fit and correlations with relevant clinical outcomes.

## Limitations

Even though the current data set can be considered as a representative sample for bimodal patients [17], the sample size is limited. The relationship between loudness growth and bimodal benefit has therefore only been tested with a qualitative approach and an explorative correlation analyses. Ideally, in a larger study group additional statistics would have been performed to identify the additional value of loudness growth above other auditory measures. Also, the newly developed loudness function was judged superior to conventional fits by visual observation, but has not been validated statistically since this is beyond the scope of the study. It is unclear how this affects results. Deviation from fit has been calculated with RMSE but is not easily compared with other literature due to different measurement conditions. Another limitation of this study is that the effect of speech band importance has not been evaluated. Loudness growth was evaluated without taking into account the relative importance of each frequency band for speech recognition. Studies have shown however, that conventional Speech Intelligibility Models (SII) are not easily applied to CI users [52]. Also, although speech outcomes were measured with CIHA, loudness was not measured with CI and HA worn simultaneously. To remediate this shortcoming, bimodal loudness was estimated by calculating the sum of loudness growth with CI and HA separately (see Fig 1). Also, no measure of localization was included. Regarding the different mechanisms of bimodal hearing, localization would theoretically profit more from loudness equilization instead of differentiation. After all, the ability to localize sound sources depends on interaural differences in time and level (ITD/ILD) which requires equivalent loudness information from both CI and HA [53]. Finally, it should be highlighted that due to the explorative nature of this study, no adjustment for multiple correlations has been applied in the statistical analysis thereby increasing the chance of finding false positive results. Ideally, and with a larger sample size, a factor analysis would have been performed to explore multi-factorial relationships.

## Future directions

Where in clinical audiology the audiogram is conventionally the commonly used parameter, the loudness audiogram might prove to be a valuable extension for evaluation of hearing loss and fitting of CI, HA and bimodal combinations. Results from this sample size limited study, suggest that loudness growth is related to bimodal benefit for multiple clinical outcome measures. Dynamic range, as estimated from loudness growth, showed similar relationships for bimodal benefit. It is undetermined which parameter is most clinically associated with outcomes. Study procedures should also be performed in an unaided setting to validate protocols without the added complexity of hearing devices. Further research should be conducted in a larger group of bimodal patients with different degrees of residual hearing to verify our results.

## Conclusions

The primary objectives of this study were to explore loudness growth differences between modalities in a group of bimodal subjects, relate loudness growth to speech recognition outcomes, and introduce the loudness audiogram. By establishing different properties of how loudness growth was combined between devices, direct comparisons with speech performance were enabled. Results show that loudness growth is related to bimodal benefit for speech recognition in noise and to some aspects of speech quality. No correlations between loudness and speech in quiet were found. Overall, we found in our study sample that subjects who had predominantly differential loudness input from the hearing aid, gained more bimodal benefit compared to those patients whose hearing aid provided mainly equivalent input.

## Supporting information

**S1 Fig. Plots (2D) of loudness growth curves measured NB and BB.** CI is shown in blue, HA in orange. Original ACALOS data are represented as dots.
(TIF)

**S2 Fig. Examples of the newly developed loudness function (red) and three existing model functions: Brand (blue) [40], Oetting (orange) [45] and Theelen-van den Hoek (green) [46].** Fits are presented for aided measurements with CI and HA for one example patient (B06).
(TIF)

**S1 Table. Loudness measured with NB and BB, expressed as percentages relative to total AUS (Total CI+HA).** IQR = Interquartile Range.
(DOCX)

**S2 Table. Loudness measured with NB, for each frequency, expressed as percentages relative to total AUS (Total CI+HA).** IQR = Interquartile Range.
(DOCX)

**S3 Table. Results of speech in quiet and speech in noise measurements for each subject (see methods section for further explanation).** IQR = Interquartile Range.
(DOCX)

**S4 Table. Dynamic range, as estimated from ACALOS$_{NB}$ data by calculating the difference between stimulus intensity levels (dB HL) at threshold (T5CU) and at the first position where maximum CU was reached.** Dynamic range was averaged across frequencies and is shown for CI, HA and as a difference score between the two devices.
(DOCX)

**S5 Table. Correlations between loudness measured NB and BB.**
(DOCX)

**S1 Dataset. Original loudness data measured with CI and HA in conditions NB and BB.**
(XLSX)

## Acknowledgments

We are thankful to prof. dr. B. Kremer (ENT, MUMC+) for his mentorship and J. Chalupper (employed by Advanced Bionics) for critically reviewing the manuscript.

## Author Contributions

**Conceptualization:** Marc van Hoof, Erwin George, Elke Devocht.

**Formal analysis:** Lars Lambriks, Marc van Hoof, Elke Devocht.

**Investigation:** Elke Devocht.

**Methodology:** Marc van Hoof, Erwin George, Elke Devocht.

**Project administration:** Elke Devocht.

**Visualization:** Lars Lambriks, Marc van Hoof.

**Writing – original draft:** Lars Lambriks.

**Writing – review & editing:** Lars Lambriks, Marc van Hoof, Erwin George, Elke Devocht.

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
