## [Decision Letter · Decision Letter 0]

27 Dec 2022

PONE-D-22-29084Extending the audiogram with loudness growth: revealing complementarity in bimodal aidingPLOS ONE

Dear Dr. Lambriks,

Thank you for submitting your manuscript to PLOS ONE. After careful consideration, we feel that it has merit but does not fully meet PLOS ONE’s publication criteria as it currently stands. Therefore, we invite you to submit a revised version of the manuscript that addresses the points raised during the review process.

We look forward to receiving your revised manuscript.

Kind regards,

Prashanth Prabhu

Academic Editor

PLOS ONE

Journal Requirements:

Additional Editor Comments:

The use of an audiogram with loudness growth will add value to the process of aiding. The research study attempts to estimate the effect of loudness growth to speech perception, listening effort and speech quality and hence to ”reveal the complementarity in bimodal aiding.” The authors have measured the loudness function in novel ways- use of both narrow band and broadband stimuli to find which could be used as shorter method, and - Use of the spline application for interpolation of loudness growth. They have constructed a loudness audiogram has been constructed in colour which is coded for CU. The use of“overlap” in the loudness growth in CI and HA show promise in future applications. Their findings also explain the presence of bimodal benefitwhen loudness was not always matched.

Correlation of these to speech intelligibility in spatial noise, ease of listening and sound quality evaluate the clinical utility of loudness function.

A description of different patterns of loudness growth is a welcome addition. These are noteworthy and will add value to both clinical work and research. Statistical analyses are appropriate. These are the strengths.

The paper is long. Ease of reading may be better if the aims are better defined. The authors have measuredbimodal benefit using speech intelligibility at different SNRs, quality of speech, which is mentioned much later.

Clarity needs to be improved.- I have added some excerpts from the text to illustrate.

1.Aim in abstract: Clinically, the audiogram is the most commonly used measure when evaluating hearing loss and fitting hearing aids. As an extension, we present the loudness audiogram, which does not only show auditory thresholds but also visualises the full course of loudness perception.

2.Aim –lines 145-150 -The main goal of this study was to measure, visualize and compare loudness growth with both modalities (CI and HA) in a group of bimodal subjects and explore the relationship between bimodal loudness growth and bimodal speech performance. As a subgoal, to evaluate whether loudness growth provided distinctive information compared to threshold estimation alone, loudness was compared with audiometric thresholds and dynamic range. Also, to assess if measuring time for loudness scaling can be reduced, results of loudness growth measured with broadband and narrowband stimuli were compared

There are some parts that are slightly redundant- For e.g .Line 235 mentions functions evaluated- but line 250 gives a better description and may be given in the beginning of the methods section. A flow chart would add to ease of reading. A conclusion may be added to the abstract.

Introduction needs to be more focused and shortened.

I have some queries which follow.

Methods: Loudness scaling was carried out with the device in the ear as mentioned in line 166. .“All measurements were performed in a sound attenuated booth with subjects using their own hearing devices at daily use settings with no manipulations allowed during testing.” The device was used in the everyday condition.How was bimodal fitting achieved? Were the devices matched for loudness ?In hearing aids, the compression is set already, so it would have affected the loudness scaling. What was the prescription method used? Those settings need to be given. I notice that loudness for 250 Hz continues to be steep and may possiblybe explained by those settings, and possibly the hearing thresholds of the participants, which show a great variability.-Figure 1 and as seen in Devocht et al.( 2017).It may also expected, that the loudness with CI would never reach very loud because of the setting of the maximum level of stimulation.

In the light of the previous statement, the lines in 203-207 are confusing. “Both hearing aids and cochlear implants can alter the shape of the loudness curve due to individual settings such as compression rules and output limiting functionalities [45,46]. Therefore, in this study a newly developed loudness function was introduced. It aims for higher accuracy and less bias compared to current fitting functions when loudness scaling is performed in free field using hearing aids or cochlear implants. A detailed step-by-step description of the fit is shown in Table 1. The loudness function was programmed in Mathematica 12.3 (Wolfram Research, Champaign, USA)”

Results may be presented in terms of the aims. The order of presentation may be mentioned.Table 1 is nicely designed with the function and figures put together.However the figures inserted in the table seem incomplete. -e.g.Table1B1p2.tif does not have any markings on the axis or abscissa and does not give a complete picture inspite of the explanation-“values of all cases where Y, loudness perception CU is zero”. Section D- Area under the surface calculation also needs a result

Speech recognition, speech understanding and “speech” have been used synonymously –speech recognition is more appropriate

Line 235-line 250 similar, better defined in 235ne 240 ‘ Threshold is misspelt

Line 285- Median ratio’s of loudness- does not need an apostrophe

Tables must match format of the journal. Tables 2,3, 4 have a very long titles. Can some part be added as a footnote? Value of N number of subjects may be indicated in the table and reasons added in the footnote.

In the tables, commas are used instead of a point to indicate probability or Rho values.If space permits, participant details may be given.

Reviewers' comments:

Reviewer's Responses to Questions

**Comments to the Author**

1. Is the manuscript technically sound, and do the data support the conclusions?

Reviewer #1: Yes

2. Has the statistical analysis been performed appropriately and rigorously? 

Reviewer #1: Yes

3. Have the authors made all data underlying the findings in their manuscript fully available?

Reviewer #1: Yes

4. Is the manuscript presented in an intelligible fashion and written in standard English?

Reviewer #1: Yes

5. Review Comments to the Author

Reviewer #1: The topic of the paper is definitely interesting for readers. Although the number of participants is relatively small, the study was performed in a rigid and reliable manner. The authors have looked into new method of bimodal fitting and introduced the concept of loudness audiogram.

Minor Comments

1. Standard deviation of age of the participants needs to be mentioned

2. More information on the aided performance participants on the cochlear implant side

3. Information on details of programming of hearing aids/gain settings

6. PLOS authors have the option to publish the peer review history of their article (what does this mean?). If published, this will include your full peer review and any attached files.

Reviewer #1: **Yes: **Saravanan Palani

---

## [Author Response · Author response to Decision Letter 0]

20 Jan 2023

We would like to thank the reviewers for their time, effort and interest in reviewing the manuscript. The noted comments provided a valuable contribution to the revised version of the manuscript. A point-to-point reaction of the authors to the reviewers comments can be found in the revised submission.

With kind regards,

Lars Lambriks

---

## [Decision Letter · Decision Letter 1]

7 Mar 2023

PONE-D-22-29084R1Extending the audiogram with loudness growth: revealing complementarity in bimodal aidingPLOS ONE

Dear Dr. Lambriks,

Thank you for submitting your manuscript to PLOS ONE. After careful consideration, we feel that it has merit but does not fully meet PLOS ONE’s publication criteria as it currently stands. Therefore, we invite you to submit a revised version of the manuscript that addresses the points raised during the review process.

We look forward to receiving your revised manuscript.

Kind regards,

Prashanth Prabhu

Academic Editor

PLOS ONE

Journal Requirements:

Reviewers' comments:

Reviewer's Responses to Questions

**Comments to the Author**

1. If the authors have adequately addressed your comments raised in a previous round of review and you feel that this manuscript is now acceptable for publication, you may indicate that here to bypass the “Comments to the Author” section, enter your conflict of interest statement in the “Confidential to Editor” section, and submit your "Accept" recommendation.

Reviewer #2: All comments have been addressed

2. Is the manuscript technically sound, and do the data support the conclusions?

Reviewer #2: Yes

3. Has the statistical analysis been performed appropriately and rigorously? 

Reviewer #2: Yes

4. Have the authors made all data underlying the findings in their manuscript fully available?

Reviewer #2: Yes

5. Is the manuscript presented in an intelligible fashion and written in standard English?

Reviewer #2: Yes

6. Review Comments to the Author

Reviewer #2: Review for:

Extending the audiogram with loudness growth: revealing complementarity 1 in bimodal aiding

Submitted to PLOS ONE

Manuscript Number: PONE-D-22-20984-R1

Comments to the associate editor

Thank you for the opportunity to review this revised article. The motivation for the study is convincing and has good clinical implication. The paper has good writing. If revised with minor changes, such a paper will add value to the existing literature. With this intent, my review here attempts to highlight some of the minor issues and indicate the errors that are reflected in writing. I sincerely hope that the authors find this helpful in refining their manuscript.

Summary

The major objectives of this study were to obtain loudness audiogram, find the relationship between loudness growth and speech recognition in quiet, in noise and speech quality in CI alone, hearing aid alone, and CI+HA conditions, and to find out whether equal input from CI and HA leads to better speech recognition or not. The study included 15 bimodal users. Loudness growth was measured with the cochlear implant and 20 hearing aid separately using a loudness scaling procedure. The results showed that loudness growth was related to bimodal benefit for speech recognition in noise and some aspects of speech quality. Gain/mapping settings leading to best speech recognition are better than setting giving equal input from both devices.

Overall Impression (General Comments)

The strength of the manuscript is the motivation of the study and the flow in writing. The introduction section is adequate. The participants, materials, and procedures are explained well. However, the main drawback of the study is the small number of participants.

Detailed review

• Title: In my opinion, the title needs more clarity

Abstract:

• Well written

Main script:

Introduction: The introduction is clear, and the objectives are well-written. Some minor changes may be required. They are given below:

o The term ‘participants’ can be used in place of ‘subjects’ though out the manuscript

o In line 71, ‘numbers of patients’ should be changed to ‘number of patients’

o In line nos. 80 and 82, ‘wearing a CI and a conventional hearing aid in opposite ears’ should be modified to ‘wearing a CI in one ear and a conventional hearing aid in the opposite ear’

o Justification for adding sound quality measurement is not given in the introduction

o Though introduction has clarity, justification for all the objectives need to be given, eg., for the last objective, ‘Evaluate if measuring time of loudness scaling can be reduced by using broadband instead of narrowband stimuli’

o The study involves adult listeners. But in the introduction, it is not clear whether the reviewed studies included adults or children.

Method:

• The method section explains the procedure well

Results and discussion:

• The results are explained in detail, and all the points are discussed adequately

• However, the result section is too lengthy

Figures:

• Could not view figure 3 in the manuscript

7. PLOS authors have the option to publish the peer review history of their article (what does this mean?). If published, this will include your full peer review and any attached files.

Reviewer #2: No

---

## [Author Response · Author response to Decision Letter 1]

30 Mar 2023

• In my opinion, the title needs more clarity

We have changed the title to:

“Extending the audiogram with loudness growth: the complementarity of electric and acoustic hearing in bimodal patients”

• The term ‘participants’ can be used in place of ‘subjects’ though out the manuscript

As we believe the term ‘subjects’ fits better within the context of this article, we have not incorporated this comment. 

• In line 71, ‘numbers of patients’ should be changed to ‘number of patients’

Has been revised accordingly. 

•In line nos. 80 and 82, ‘wearing a CI and a conventional hearing aid in opposite ears’ should be modified to ‘wearing a CI in one ear and a conventional hearing aid in the opposite ear’.

Has been revised accordingly.

•Justification for adding sound quality measurement is not given in the introduction

In line 91, we have now added:

‘Also, sound quality, which is often perceived as unnatural with CI alone [23], improved by adding a contralateral HA.’

•Though introduction has clarity, justification for all the objectives need to be given, eg., for the last objective, ‘Evaluate if measuring time of loudness scaling can be reduced by using broadband instead of narrowband stimuli’.

We have now rephrased line 141:

“Evaluate if measuring time of loudness scaling can be reduced by using broadband instead of narrowband stimuli, thereby reducing the amount of tests within the procedure.”

We believe the other study objectives are either self-evident or are sufficiently introduced within the introduction. 

• The study involves adult listeners. But in the introduction, it is not clear whether the reviewed studies included adults or children.

Has been revised accordingly. 

• The result section is too lengthy

We have shortened the results section by removing sections of the qualitative analysis and text within ‘Loudness and bimodal benefit’. If the reviewer believes the results section is still too lengthy, we could consider removing sections to the supplementary material. 

• Could not view figure 3 in the manuscript

Figure 3 has been added in the original submission. Perhaps there were errors due to file size restrictions in the manuscript preparation. We have therefore also added Figure 3 below (in the response to the reviewers file).

---

## [Editor Report · Decision Letter 2]

3 Apr 2023

Extending the audiogram with loudness growth: the complementarity of electric and acoustic hearing in bimodal patients

PONE-D-22-29084R2

Dear Dr. Lambriks,

We’re pleased to inform you that your manuscript has been judged scientifically suitable for publication and will be formally accepted for publication once it meets all outstanding technical requirements.

Kind regards,

Prashanth Prabhu

Academic Editor

PLOS ONE
---

## [Editor Report · Acceptance letter]

10 Apr 2023

PONE-D-22-29084R2 

Extending the audiogram with loudness growth: the complementarity of electric and acoustic hearing in bimodal patients 

Dear Dr. Lambriks:

I'm pleased to inform you that your manuscript has been deemed suitable for publication in PLOS ONE. Congratulations! Your manuscript is now with our production department. 

Kind regards, 

on behalf of

Dr. Prashanth Prabhu 

Academic Editor

PLOS ONE